Phytocenosis biodiversity at various water levels in mesotrophic Lake Arakhley, Lake Baikal basin, Russia

Tsybekmitova Gazhit Ts. gazhit@bk.ru 1
Radnaeva Larisa D. 2
Tashlykova Natalya A. NatTash2005@yandex.ru 1
Shiretorova Valentina G. 2
Bazarova Balgit B. 1
Tulokhonov Arnold K. 2
Matveeva Marina O. 1
1 Laboratory of Aquatic Ecosystem, Institute of Natural Resources, Ecology and Cryology of the Siberian Branch of the Russian Academy of Sciences , Chita , Zabaykalskii krai , Russian Federation
2 Laboratory of Chemistry of Natural Systems, Baikal Institute of Nature Management of the Siberian Branch of the Russian Academy of Sciences , Ulan-Ude , Buryatia , Russian Federation
Christoffersen Kirsten
Electronic publication date: 2021 Jun 18
Publication date: 2021
Volume: 9
Electronic Location ID: e11628
Received 2020 Oct 21; Accepted 2021 May 26
Copyright: ©2021 Tsybekmitova et al.
Copyright year: 2021
Copyright holder: Tsybekmitova et al.
License: This is an open access article distributed under the terms of the Creative Commons Attribution License, which permits unrestricted use, distribution, reproduction and adaptation in any medium and for any purpose provided that it is properly attributed. For attribution, the original author(s), title, publication source (PeerJ) and either DOI or URL of the article must be cited.
License URL: https://creativecommons.org/licenses/by/4.0/

Keywords: Freshwater lake, Level regime of lake, Abiotic factors, Phytoplankton, Hydrophytes, Redundancy analysis (RDA)

Funding: Federal research programs project FUFR-2021-0006 project FUFR-0273-2021-0004 Russian Foundation for Basic Research 17-29-05085 ofi_m This work was supported by federal research programs (project FUFR-2021-0006 and project FUFR-0273-2021-0004) and the Russian Foundation for Basic Research, grant no. 17-29-05085 ofi_m. The funders had no role in study design, data collection and analysis, decision to publish, or preparation of the manuscript.

==============================
Small lakes have lower water levels during dry years as was the case in 2000–2020. We sought to show the biodiversity of plant communities at various water levels in Lake Arakhley. Changes in moisture content are reflected in the cyclical variations of the water level in the lake, which decreased approximately 2 m in 2017–2018. These variations affect the biological diversity of the aquatic ecosystems. We present the latest data on the state of the plant communities in this mesotrophic lake located in the drainage basin of Lake Baikal. Lake Arakhley is a freshwater lake with low mineral content and a sodium hydrocarbonate chemical composition. Changes in the nutrient concentration were related to precipitation; inflow volume and organic matter were autochtonous at low water levels. The most diverse groups of phytoplankton found in the lake were Bacillariophyta, Chlorophyta, and Chrysophyta. High biodiversity values indicate the complexity and richness of the lake’s phytoplankton community. A prevalence of Lindavia comta was observed when water levels were low and Asterionella formosa dominated in high-water years. The maximum growth depth of lacustrine vegetation decreased from 11.0 m to 4.0 m from 1967 to 2018. Decreasing water levels were accompanied by a reduction in the littoral zone, altering the communities of aquatic plants. The hydrophyte communities were monodominant in the dry years and were represented by Ceratophyllum demersum. The vegetation cover of the lake was more diverse in high-water years and variations in the lake’s water content altered the composition of biogenic substances. These changes were reflected in the lake’s phytocenosis.

Introduction

The global nature of climate change has been widely discussed (Koch, 2001; Riis & Hawes, 2002; Bornette & Puijalon, 2011; Ji et al., 2014; Donchyts et al., 2016; Huang et al., 2016) and climatic changes can be detected at the regional level when we examine environmental factors such as level of the water surface, the mode of its fluctuations and the temperature of the atmospheric air (Hill, Keddy & Wisheu, 1998; Obyazov, Usmanov & Zhilin, 2002; Obyazov, 2011; Kuang & Jiao, 2016; Yang et al., 2016; Yang et al., 2014b).

The ecological effects of climate change are most evident in primary aquatic production processes (Paerl et al., 2003). The filling and evaporation of lakes due to climatic fluctuations indirectly regulates the growth and development of plant communities (Bornette & Puijalon, 2011). Phytoplankton dynamics are closely related to annual fluctuations in temperature and water levels, the mixing of the water column, and the availability of nutrient resources and their consumption (Winder & Sommer, 2012). Phytoplankton structure, seasonal dynamics, and taxonomic composition are directly or indirectly affected by these environmental factors due to climate changes (Winder & Sommer, 2012; Liu et al., 2015; Izaguirre et al., 2015). These factors influence various physiological processes, causing a shift in the timing and extent of plankton algal blooms and favor the development of species that are most adapted to changing climatic conditions (Nõges, Nõges & Laugaste, 2003; Adrian, Wilhelm & Gerten, 2006; Lewandowska & Sommer, 2010; Zohary, Flaim & Sommer, 2020).

Recent studies have focused on the relationship between the structural characteristics of phytoplankton and environmental factors (Bierman & Dolan, 1981; Watson, McCauley & Downing, 1997; Nalewajko & Murphy, 2001; Tian et al., 2013; Chang et al., 2013; Wang et al., 2015). These studies focus on the factors with the greatest impact on phytoplankton communities. According to this research, the structural characteristics of phytoplankton are regulated by temperature, illumination, nutrients, and water level (Yang et al., 2016; Yang et al., 2014b; Yang et al., 2017).

Macrophytes are reliable indicators of long-term changes in the littoral zone, but they do not reflect rapid changes in water quality (Palmer, Bel & Butterfield, 1992; Melzer, 1999). Macrophytic thickets are the main trophic community and are a natural biofilter for substances entering the ecosystem. Recent studies have found decreasing aquatic vegetation in approximately 65.2% of the world’s lakes (Zhang et al., 2017).

It is important to monitor and analyze water ecosystems to improve our awareness of the possible negative biological effects caused by climate changes and to regulate anthropogenic activities. Research on the biological diversity of the organizational levels and dynamics in aquatic ecosystems to the north of Central Asia may contribute to the understanding of biosphere balance, especially in water ecosystems.

The purpose of this study is to assess the biodiversity of plant communities under changing water levels in Lake Arakhley.

Materials and Methods

Study Area

Lake Arakhley is located south of the Vitim Plateau within the Beklemishev tectonic basin. The area is characterized by a continental subarctic climate with considerate diurnal temperatures, long, cold, dry winters and short, hot summers with more precipitation occurring during the latter half of the season. The lake is used for recreation and fishery purposes and is approximately 100 km from Chita, the capital of the Zabaikalsky region (Fig. 1, Table 1).

Lake Arakhley’s water regime is determined by changes in the ratio of water balance elements. Precipitation on the reservoir surface is slightly higher than the inflow (precipitation equals 17.8 million m3/year; inflow equals 16.1 million m3/year) and accounts for the incoming water balance of Lake Arakhley. Water typically dissipates through evaporation (27.1 million m3/year) and runoff from the lake only accounts for 20% of the outflow (Obyazov, 2011). Consequently, changes in the moisture content are reflected in the cyclical variations in the water level. The cyclical variations of the water level coincide with the cyclicality of the atmospheric precipitation. The integral difference curve reveals the phases of low (precipitation below normal) and high (precipitation above normal) moisture content of the territory (Obyazov, 2011). Long-term changes in Lake Arakhley’s water levels are shown in Fig. 2.

Figure 1 Schematic map of the Arakhley Lake.

Table 1 The main characteristics of the Arakhley Lake (Obyazov, Usmanov & Zhilin, 2002).

Characteristic, units	Arakhley	
geographical coordinates	52°48′–52°10′N, 112°45′–113°04′E	
basin bottom elevation	940-930 m BS	
surface area, km2	59.0	
water volume, km3	0.60	
length, km	11.0	
max width, km	6.7	
average width, km	5.3	
max depth, m	17.0	
average depth, m	10.2	
shore length, m	29.0	
catchment area, km2	256	
water exchange coefficient	0.055	
trophic status	mesotrophic	
Notes.

BS from zero mark of Baltic height system

Figure 2 Integral difference curve of annual amounts of atmospheric precipitation (Obyazov, 2011) and changes in the water level of the Lake Arakhley. Hydrological post of Preobrazhenka village. Post zero mark: 963.1 m (Baltic height system: BS).

The water level was highest in 1962 and decreased over the following year. The decline in the water level slowed between 1967–1968 then increased until 1972. The next low-water phase continued until 1980, at which point the water level increased over the next 4–6 years. Water level fluctuations continued until 1990–1991. In 2008 the water level was close to the absolute minimum (Obyazov, 2011; Obyazov, 2014). We observed the minimum water level in 2017–2018, when the level decreased by approximately 2 m. This decrease caused a 50 m displacement of the water’s edge, causing the littoral zone, composed of sand and gravel deposits, to disappear.

Physical and chemical data

We used data from 2011, 2013, 2014, and 2016–2018 to analyze the long-term dynamics of hydrochemical parameters (Tsybekmitova & Subbotina, 2013; Tsybekmitova et al., 2020). Hydrochemical data taken during the low-water period in Lake Arakhley show the state of its ecosystem (Fig. 2). Samples were collected from the surface and bottom layers using a Schindler-Patalas trap (PB-6, Borok, Russia). Surface horizon samples were taken 0.10 m below the water surface level, and bottom water samples were taken 0.10 m above the bottom. We conducted a chemical analysis of the water to identify nitrites using Griss reagent, nitrates (reduction to nitrites with the Griss reagent), ammonium ions (using the Nessler reagent), phosphates (with a reagent mixed with ascorbic acid), and total phosphorus (by combustion with potassium persulfate). The organic carbon concentration was determined by adding wet potassium dichromate oxidation (COD) and permanganate oxidation (PO) to lake water samples. These methods are described in detail by Adamovich et al. (2019). Concentrations were determined using a Spekol-1300 spectrophotometer. We used the ion chromatography system ICS-1600 and a Profile Plus inductively coupled plasma atomic emission spectrometer to determine the macro- and micro-components of water with 2% accuracy. The physical and chemical parameters of water (pH, total dissolved solids, water temperature, electrical conductivity, oxidation–reduction potential, salinity, dissolved oxygen content, and turbidity) during sampling were measured using AQWA-READER (Germany). These methods have been used since 2000 to monitor Lake Arakhley long term.

Phytoplankton data

Phytoplankton samples were taken in May–June, July–August, October, and December of 2017–2018. We used data from the following periods to analyze the long-term dynamics of phytoplankton: 1966–1969 (Morozova, 1975; Morozova, 1981), 1970–1971 (Morozova & Shishkin, 1973; Morozova, 1981), 1988 and 1994–1995 (Ogly, 1993; Ogly, 1995), and 2008–2009 (Tashlykova & Koryakina, 2013).

Phytoplankton was collected from 0.5–1.0 L water samples taken from two or three horizons (surface, transparency, bottom) using the Schindler-Patalas sampler (PB-6, Borok, Russia) in the deep-water sites of the lake. Quantitative samples were fixed with a 4% formalin solution and quality samples were fixed with Lugol’s solution. The samples were prepared using the sedimentary method and each sample was processed separately. Algae were counted according to the Hansen method (Sadchikov, 2003) on a counting plate. The biomass was determined based on the volume of individual algae cells or colonies and their geometric figures. The specific weight was taken equal to one unit. Abundance and biomass were calculated as a weighted arithmetic mean (Sadchikov, 2003). Taxon classification and algae group synonymy were taken from the algological site AlgaeBase (Guiry and Guiry ©1996–2020). Species diversity was calculated using Shannon’s index (Shanon & Weaver, 1963). The evenness or equitability index was calculated using Pielou’s formula (Pielou, 1967).

Macrophytes data

Water plants were studied using their ecological profiles (Yunnatov, 1964; Katanskaya, 1981), which reflected the distribution of aquatic phytocenoses at specific depths. Ten meter-wide profiles were observed from the water edge to the maximum depth at which the plant grew. Our observations included registering the species diversity, selecting the phytomass of plants, and recording the depth of the water and the nature of the soil. Plants were collected from the reservoir using a cat anchor with a metal mesh, which could also collect small plants. The cat anchor was cast along the depth change gradient five to 15 times, depending on the density of the communities. We studied eight profiles around the lake, covering the entire coastal area. Plant phytomass was selected using a device for the quantitative accounting of gammarids (KUG) with a capture area of 0.25 m2 (Bazarova & Itigilova, 2006). The phytomass data are given in their absolute dry value. We listed the dominant species sequentially to create a graphical description of the composition and structure of vegetation in the ecological series with increasing depths. In the ecological row of macrophytes unidirectional changes in communities are indicated by single-pointed arrows. Double-pointed arrows indicate moving species; plus signs indicate species forming communities at the same depth.

Data analysis

We used Microsoft Excel and XLSTAT (Addinsoft, USA) to conduct the statistical and mathematical analyses of the findings. Redundancy analysis was performed to illustrate the correlation between the composition of the plankton and abiotic factors (Table 2). The absolute value of the load above 0.90 was considered to be significant.

The data were normalized by dividing the initial data by the standard deviation of the corresponding variables (Shipunov et al., 2014).

Results

Physical and chemical parameters of the lake ecosystem

The physical and chemical parameters of environmental waters depend on the dissolution and chemical weathering of rocks and biogeochemical processes in the soils of the catchment area and bottom sediments of water bodies. We show the specific physical and chemical parameters of the lowest water level (Fig. 2) in Table 3.

The temperature exchange between the water, atmosphere, and bottom sediments resulted in stratified thermal conditions in 2017 and 2018 with a gradient slightly above 3 °C between the surface and bottom layers (Table 3). The lake water was slightly alkaline. The oxidation–reduction potential (ORP) of the ecosystem in Lake Arakhley ranged from 200 to 218 mV, creating oxidative conditions and a number of elements in their highest valence. The ORP values of the surface and bottom horizons of the water column differed slightly due to more favorable oxygen conditions in the surface water layer. The oxygen concentration in the surface horizon was higher (11.4 mg O2 L−1) compared to the bottom layers (8.8 mg O2 L−1). There was a slight difference between the parameters of the surface and bottom water layers in terms of TDS and EC. Lake Arakhley is a low-mineralized freshwater lake with regard to total dissolved solids. We determined the main chemical composition and concentration values of specific elements (Table 3) and found that the water was of the hydrocarbonate class with a calcium-sodium group. The contrast analysis of macro- and microelement compositions indicated that the values did not exceed MPC. However, manganese concentration was 86 times higher than the MPC in the bottom water layers (Table 3).

Table 2 Parameters and their abbreviations.

Description of a plankton	Abbreviation	Description of abiotic parameters	Abbreviation	
number	n	lake depth	H	
phytoplankton species	ph	water transparency	TR	
Cyanobacteria	cya	pH	pH	
Bacillariophyta	bac	total dissolved solids	TDS	
Chrysophyta	chr	water temperature	T	
Charophyta	cha	electrical conductivity	EC	
Chlorophyta	chl	oxidation–reduction potential	ORP	
Euglenophyta	eug	salinity	Sal	
Dinophyta	din	turbidity	Turb	
total number of	N	nitrites	NO2	
total biomass of	B	nitrates	NO3	
		ammonium	NH4	
		phosphates	PO4	
		permanganate oxidability index	PI	
		chemical oxygen demand	COD	
		chlorophyll a concentration	Xla	

Biogenic elements in the lake during the most productive summer period are shown in Table 4.

The nitrate form of nitrogen was dominant during the summer period. The nitrites and ammonium ions were minimal. The concentration of phosphorus was greater in the bottom layers compared to the surface layers. The organic matter was of autochthonous origin in the PO/COD ratio.

The analysis of year-to-year variations of biogenic elements showed that the ammonium and nitrite-nitrogen concentrations declined from 2011 to the present day, while the concentration of nitrates increased (Fig. 3).

According to the results from 2011–2018, the PO/COD ratio was below 50%, which indicates the autochthonous origin of the organic matter. The concentration of organic matter resistant to oxidation (COD) increased since 2013, but has not reached the level of 2011. The content of easily-oxidized organic substances (PO) is consistent according to the data of 2011–2018 (Fig. 4).

Table 3 Average annual characteristics physical and chemical parameters in the central zone of the Lake Arakhley for the period open water between 2017 and 2018.

Physical and chemical parameters	Water horizon	MPC*	
	surface layer (n = 24)	bottom layer (n = 24)		
T, °C	12.2 ± 2.3	8.8 ± 0.8	–	
pH	7.8 ± 0.2	7.8 ± 0.1	6.5–8.5	
ORP, mV	218 ± 21.5	200 ± 23.3	–	
EC, µS cm−1	230 ± 6.7	226 ± 8.6	–	
TDS, mg L−1	153 ± 7.0	146 ± 6.3	1000	
DOC, mg O2 L−1	11.4 ± 0.8	8.8 ± 0.9	<4.0	
TURB, NTU	36.2 ± 2.4	36.2 ± 2.4	–	
Sal, g kg−1	0.11 ± 0.003	0.11 ± 0.003	–	
Cl−, mg L−1	2.08 ± 0.43	2.23 ± 0.46	300	
SO42−, mg L−1	1.15 ± 0.08	1.03 ± 0.07	100	
HCO3−, mg L−1	136.64 ± 3.15	142.74 ± 2.43	–	
Na+, mg L−1	15.46 ± 1.23	16.65 ± 0.99	120	
K+, mg L−1	3.5 ± 0.16	3.6 ± 0.19	50	
Mg2+, mg L−1	7.04 ± 0.55	7.01 ± 0.54	40	
Ca2+, mg L−1	20.01 ± 0.51	21.01 ± 0.56	180	
Fe, mg L−1	0.005 ± 0.002	0.019 ± 0.013	0.1	
Zn, mg L−1	0.0007 ± 0.0004	0.0038 ± 0.0023	0.01	
Mn, mg L−1	< 0.001	0.865 ± 0.099	0.01	
Pb, mg L−1	< 0.001	0.0017 ± 0.0005	0.006	
Ni, mg L−1	< 0.001	< 0.001	0.01	
Cd, mg L−1	< 00001	< 0.0001	0.005	
Cr, mg L−1	< 0.001	< 0.001	0.02	
Notes.

* MPC: the maximum permissible concentration for lakes fishery use in Russia (FAF, 2020).

The analysis of year-to-year variations of the concentration of phosphates and total phosphorus showed that their concentrations decreased in 2013–2018 compared to 2011 (Fig. 4).

Table 4 The concentration of nutrients (nitrogen and phosphorus) and organic matter content (OMC) in the central zone of the Lake Arakhley for the summer period 2017–2018 (mg L−1).

Water horizon	Nitrogen	Phosphorus	OMC	
	NO2−	NO3−	NH4+	PO43−	Ptotal	PO	COD	
surface layer (n = 25)	0.001	0.51	0.004	0.014	0.031	4.89	10.28	
SD	0.0001	0.048	0.0004	0.0013	0.0011	0.97	3.23	
bottom layer (n = 25)	0.006	0.56	0.002	0.023	0.038	6.1	12.43	
SD	0.0007	0.054	0.0002	0.0022	0.0022	6.2	4.1	
Notes.

PO permanganate oxidability

COD chemical oxygen demand

Phytoplankton composition

A total of 39 taxa ranking below the genus level were detected in the lake’s phytoplankton sampled from the central zone between 2017–2018 (Table 5).

There were 97 algae taxa recorded from the lake’s center and coastal areas during the spring-to-summer period of 2017 (Tashlykova, 2019). The most diverse groups were the diatoms (Bacillariophyta), green algae (Chlorophyta), and golden algae (Chrysophyta) comprising 79.9% of the total identified taxa. Among the most frequently identified species were the following: Lindavia comta (Kützing) Nakov, Gullory, Julius, Theriot & Alverson, Fragilaria crotonensis Kitton, F. radians (Kützing) D.M.Williams & Round, Ulnaria ulna (Nitzsch) Compère in Jahn et al., Nitzschia graciliformis Lange-Bertalot & Simonsen, Chrysococcus rufescens Klebs, Dinobryon cylindricum O.E. Imhof, D. divergens O.E. Imhof, D. sertularia O.E.Imhof, Kephyrion spirale (Lackey) Conrad, and Oocystis marssonii Lemmermann.

The value of the weighted average abundance and biomass is shown in Fig. 5.

Figure 3 Average annual concentrations of nitrogen compounds in the Lake Arakhley.

Figure 4 Average annual concentration of the easily oxidable organic matter and organic matter resistant to oxidation (A), phosphates and total phosphorus (B) in Lake Arakhley.

Algae’s quantitative development in the spring was insignificant when compared with the predominance of small-sized Chrysophyta. Algocenoses composition was comprised of Chrysophyta (10–75% of total taxa and 30–92% of the overall biomass) and Bacillariophyta (20–85% of total taxa and 10–60% of the overall biomass). The total number of algae and overall biomass increased in the summer by an order of magnitude due to a measurable growth of large diatoms. Bacillariophyta prevailed in the phytoplankton with 60–80% of total taxa and 50–95% of the overall biomass. The winter season was similarly marked by the predominance of Bacillariophyta in the algocenosis. We assessed the biodiversity in phytoplanktonocenoses using the Shannon and Pielu indices. The phytoplankton of the lake was characterized by high values of these indices, which demonstrates the complexity and high diversity of the community. The Shannon index varied from 2.97 to 3.11 in the spring and the Pielou index ranged from 0.52−0.76. In the summer, the value of the Shannon index was 1.27−1.38 and the Pielou index varied from 0.15 to 0.28. These values correspond to algae’s seasonal distribution. The maximum values of the Shannon index are recorded in the spring and autumn, and the minimum values are recorded in the winter and summer.

Table 5 Species composition of phytoplankton in the central part of Lake Arakhley.

No.	Taxa	2017	2018	
		May–June	July–August	October	December	May–June	July–August	October	December	
1	2	3	4	5	6	7	8	9	10	
	Cyanobacteria									
1	Gomphosphaeria lacustris Chodat, 1898	+	–	+	+	–	+	–	+	
2	Anabaena sp.	+	–	+	–	–	–	–		
3	Aphanizomenon flos-aquae Ralfs ex Bornet & Flahault, 1886	+	–	+	–	+	–	+	–	
	Bacillariophyta									
4	Lindavia comta (Kützing) Nakov, Gullory, Julius, Theriot & Alverson, 2015	+	+	+	+	+	+	+	+	
5	Melosira varians C.Agardh, 1827	–	–	–	–	–	+	–	+	
6	Fragilaria crotonensis Kitton, 1869	+	+	–	–	+	+	–	+	
7	F. radians (Kützing) D.M.Williams & Round, 1987	+	–	–	–	+	+	–	+	
8	Ulnaria ulna (Nitzsch) Compère in Jahn et al., 2001	+	–	–	–	+	–	–	+	
9	Asterionella formosa Hassall, 1850	–	+	+	–	–	+	+	+	
10	Nitzschia graciliformis Lange-Bertalot & Simonsen, 1978	+	–	+	–	–	–	–	–	
	Chrysophyta									
11	Chrysococcus rufescens Klebs, 1892	+	+	–	+	–	–	–	+	
12	Dinobryon bavaricum Imhof, 1890	+	–	–	–	+	–	–	+	
13	D. cylindricum O.E. Imhof, 1887	+	–	–	–	+	–	–	+	
14	D. divergens O.E.Imhof, 1887	+	+	+	–	+	–	+	+	
15	D. elegans Korshikov, 1926	+	–	–	–	+	+	–	–	
16	D. sertularia Ehrenberg, 1834	+	–	–	–	+	–	–	–	
17	Kephyrion spirale (Lackey) Conrad, 1939	+	–	–	–	+	–	–	+	
	Charophyta									
18	Elakatothrix genevensis (Reverdin) Hindák, 1962	+	–	–	–	–	–	–	–	
19	Cosmarium sp.	–	+	–	+	–	+	+	+	
20	Staurastrum sp.	+	–	+	+	–	+	+	+	
	Chlorophyta									
21	Mucidosphaerium pulchellum (H.C.Wood) C.Bock, Proschold & Krienitz, 2011	–	+	–	+	–	+	–	+	
22	Actinastrum hantzschii Lagerheim, 1882	+	–	–	–	–	+	–	–	
23	Oocystis marssonii Lemmermann, 1898	+	+	+	+	+	+	+	+	
24	O. borgei J.W.Snow, 1903	–	+	+	+	–	+	–	+	
25	Lagerheimia genevensis (Chodat) Chodat 1895	–	–	+	–	–	+	–	–	
26	Monoraphidium contortum (Thuret) Komárková-Legnerová in Fott, 1969	+	–	–	–	+	+	–	–	
27	M. griffithii (Berkeley) Komárková-Legnerová, 1969	+	–	+	–	–	+	–	+	
28	M. komarkovae Nygaard, 1979	+	–	+	–	–	+	–	+	
29	Coelastrum microporum Nägeli in A.Braun, 1855	–	–	+	–	–	–	–	–	
30	Tetraëdron incus (Teiling) G.M.Smith, 1926	+	+	–	–	+	–	–	–	
1	2	3	4	5	6	7	8	9	10	
31	T. minimum (A.Braun) Hansgirg, 1888	–	–	+	+	+	–	–	+	
32	Pseudopediastrum boryanum (Turpin) E.Hegewald in Buchheim et al., 2005	–	–	–	–	–	+	–	+	
33	Schroederia setigera (Schröder) Lemmermann, 1898	–	+	–	+	–	+	–	+	
34	Chlamydomonas globosa J.W.Snow, 1903	+	+	+	–	+	–	–	–	
35	Pandorina morum (O.F.Müller) Bory in J.V.Lamouroux, Bory & Deslongschamps, 1824	–	–	+	–	+	+	–	–	
36	Koliella longiseta (Vischer) Hindák, 1963	–	–	–	+	–	–	–	+	
	Dinophyta									
37	Ceratium hirundinella (O.F.Müller) Dujardin, 1841	–	+	–	–	–	+	+	–	
38	Peridinium sp.	+	+	–	–	+	+	–	–	
	Euglenophyta									
39	Trachelomonas sp.	+	–	–	–	+	–	–	–	

From 1966 to 1969, the lake’s algocenosis featured abundant Cyclotella comta Kützing, (currently Lindavia comta) (Guiry & Guiry, 2020) (Yunnatov, 1964; Morozova & Shishkin, 1973). The lake was dominated by Asterionella formosa Hassall from 1970 to 1971 (Morozova, 1981; Morozova & Shishkin, 1973), A. formosa (Ogly, 1993; Ogly, 1995) from 1990 to 1995, Puncticulata comta (Kützing) H. Hakansson (currently L. comta) from 2008 to 2009 (Tashlykova, 2018), and L. comta from 2017 to 2018 (Table 6). The aforementioned distribution of the dominant species corresponds to the alternate phases of the hydrological cycle. L. comta was dominant during the dry years, while A. formosa was prevalent in high-water years.

Figure 5 The weighted average abundance (A, ∗103 cell ∗L−1) and biomass (B, mg ∗m−3) phytoplankton in the study period.

Table 6 Dominant phytoplankton complex and average per year biomass in the Lake Arakhley in different at different periods of study.

Parameter	Research year	
	19661	19672	19683	19694	19705	19716	19887	19948	19959	200810	200911	2017–2018	
Dominant species	Cyclotella comta; Microcystis pulverea; Holopedia geminata; Chroomonas acuta; Schroederia setigera; ubrk Ankistrodesmus pseudomirabilis; species of the genus Oocystis	Asterionella formosa; Microcystis pulverea; Holopedia geminata; Chroomonas acuta; Schroederia setigera; Ankistrodesmus pseudomirabilis; species of the genus Oocystis	Asterionella formosa; Microcystis pulverea; Anabaena spiroides; Aulacoseira granulata; Fragilaria crotonensis	Gloeotrichia echinulata; Asterionella formosa; Aulacosira granulata	Cyclotella comta; Asterionella formosa; Melosira varians; species of the genus Dinobryon; Ceratiumhirundinella; Tetraëdron incus	Cyclotella comta; Chromulina sp.; Asterionella formosa; Ceratium hirundinella; Gloeotrichia echinulata; species of the genus Dinobryon	Lindavia comta ≡Cyclotella comta, Fragilaria crotonensis, F. radians, Ulnaria ulna, Nitzschia graciliformis,Chrysococcus rufescens, Dinobryon cylindricum, D. divergens, D. sertularia, Kephyrion spirale, Oocystis marssonii	
Biomass (average per year, in g m−3)	1.356	0.557	0.21	0.366	0.245	0.2	0.69	61	26.4	1.5	2.5	 0,94-1,1	
Notes.

1-4 at Morozova (1975) and Morozova (1981).

5-6 at Morozova & Shishkin (1973) and Morozova (1981).

7-8 at Ogly (1993) and Ogly (1995).

10-11: at Tashlykova & Koryakina (2013).

There is inconsistent data on the quantitative development of algae in different periods of the hydrological cycle. The average annual biomass is shown in Table 6. According to these data, the highest biomass values were recorded for 1966, 1994–1995, 2008–2009, and 2017–2018.

Macrophyte communities

There are two species of Charophyta, one species of Bryophytes, and 58 species of vascular plants recorded in Lake Arakhley. The maximum growth depth of lacustrine vegetation decreased from 11.0 m to 4.5 m from 1967 to 2018 and the wide strip of Nitella opaca (Bruz.) Ag and Fontinalis hypnoides Hartm disappeared. From 1967 to 1974, these species occupied isobaths 7.0–11.0 m. The communities of N. opaca were still observable at four to six-meter depths along the western and eastern shores of the lake from 1998 to 2000. Moss F. hypnoides was found at a depth of 4 m on the lake’s northern coast (Bazarova & Itigilova, 2006). N. opaca and F. hypnoides populations have not been registered since 2006.

The analysis of the long-term macrophyte vegetation dynamics showed that communities of nine species were dominant in 1967 (Fig. 6) and occupied approximately 44% of the littoral area. These communities were characterized by low-density thickets. In terms of the overgrowth area and phytomass value, the dominant species were Charophyta, including the deep-water species of N. opaca, and the shallow-water species of Chara arcuatofolia (Fig. 6). Decreased Ch. arcuatofolia in 1974 led to an increase in N. opaca thickets density, and the start of the active development of vascular plants. In 1998, a sharp reduction in N. opaca thickets was recorded and an increase in the phytomass of Lemna trisulca L. and Ceratophyllum demersum L. was noted. Five species of macrophytes dominated the vegetation from 2000–2008. Species diversity was preserved from 2017–2018 with C. demersum as the dominant vegetation. Plant communities grew from the water’s edge and were represented by Potamogeton perfoliatus L., Potamogeton pectinatus (L.), and Myriophyllum sibiricum Kom. Chara arcuatofolia Vilh populations were found at depths of of 1.5–2.0 m and C. demersum grew at greater depths. L. trisulca populations were found in depressions on the lake bottom at depths of 3.0 m, whereas Potamogeton praelongus Wulf was observed at the maximum depths of plant growth (4.5 m). The abundance of M. sibiricum communities indicated an increased eutrophicity of the littoral zone. During this period, splash water bodies were formed on the coastal areas of the lake with Ch. arcuatofolia (Kuklin & Bazarova, 2019). A slight rise in the water level in 2018 did not cause significant changes in the vegetation. However, the flooding of the coastal strip initiated the development of helophyte communities represented by the thinned strip of Scrispus sp.

Figure 6 Long-term dynamics of macrophytes (by phytomass value) in Lake Arakhley.

Maximum plant germination occurred at a depth of approximately 4.5 m, which corresponded to the sixth belt of growth in 1998–2000. In the high-water period of 1967–1974, the ecological row of macrophytes in Lake Arakhley was characterized by the following sequence: Persicaria amphibia (L) Gray (depths ≈ 0.2−0.8 m) → P. perfoliatuis (≈ 1.0−1.5 m) →Ch. arcuatofolia (≈ 2.0 m) →L. trisulca ↔C. demersum ↔P. praelongus (≈ 3.0−7.0) →N. opaca ↔F. hypnoides. (≈ 7.0–11.0). In the period of 1998–2000, this row decreased but maintained its general structure: P. amphybia (≈ 0.5 m) →P. perfoliatuis (≈ 1.0 m) →Ch. arcuatofolia (≈ 2.0 m) →L. trisulca (≈ 3.0 m) ↔C. demersum (≈ 4.0 m) ↔P. praelongus (≈ 5.0 m) →N. opaca (≈ 6.0 m). Observations made in 2017–2018 revealed a decrease in the depth of plant growth and a violation of the clearly-expressed belt structure of vegetation. The row was featured the following species: P. perfoliatuis + P. pectinatus + M. sibiricum (≈ 0.1−0.8 m) → Ch. arcuatofolia (≈ 1.8 m) → C. demersum + P. praelongus (≈ 2.5–4.5 m). During the low-water period of 2017-2018, there was no clear correlation to the depths and the ecological series was as follows: P. perfoliatuis + P. pectinatus + M. sibiricum (0.1−0.8 m), Ch. arcuatofolia (1.8 m) → C. demersum + P. praelongus (2.5−4.5 m). Consequently, the lake featured a complex mosaic vegetation structure that varied along a depth gradient. Helophytes communities only grow along the southwestern coast, whereas neystophytes are represented by a narrow strip of P. amphibia along the southern and western coast with a well-developed hydatophyte community.

Phytoplankton composition in correlation with abiotic environmental factors

We sought to identify the key factors that determined the change in the structural parameters of phytoplankton in the central zone of Lake Arakhley from 2018–2019 (Tsybekmitova et al., 2020). We performed redundancy analysis and the first two components were selected, explaining 94.92% of the total variance (Fig. 7).

Figure 7 RDA analysis of phytoplankton composition and abiotic factors within the first two major factors in the central zone of the Arakhley Lake in 2017–2018 (Tsybekmitova et al., 2020).

The number of Chlorophyta and the biomass of Dinophyta showed a clear negative correlation. The number and biomass of Euglenophyta, the number of Dinophyta, the biomass of Cyanobacteria and Bacillariophyta, and the number of taxa of Cyanobacteria, Bacillariophyta, Chrysophyta, and Chlorophyta were positively correlated with the first major factor (RDA axis 1). These results explained a 74.12% variability of the community. The second major factor (RDA axis 2) explained a 20.80% variability of the pelagial communities. The number of microalgae and groups of Chlorophyta, Charophyta, Cyanobacteria, and Bacillariophyta demonstrated positive correlation with RDA axis 2. Abiotic factors had the most meaningful effect on biocenosis and the physical and chemical parameters of water (TDS, pH, EC, ORP, Turb and Sal) positively correlated with RDA axis 1. The biogenic elements (nitrites, ammonium, phosphates) negatively correlated with RDA axis 1. Depth and transparency were positively correlated with RDA axis 2.

Discussion

Significant global climate shifts in recent decades are reflected in the amount of atmospheric precipitation. The long-term global warming trends have been demonstrated by a number of studies (Ji et al., 2014; Huang et al., 2016). Rising global temperatures have resulted in climatologically wet regions becoming wetter and dry regions becoming drier. However, these changes differ from the global perspective at the regional level. Our study area in the eastern sector of Central Asia is a monsoon type with annual atmospheric precipitation. Small climate cycles appear at 24–25 years and six-to-seven years within a 60-year cycle (Obyazov, 2011; Obyazov, 2014).

Water level fluctuations are also cyclical and depend on long-term changes in atmospheric precipitation. Research on the impact of climate change on the state of aquatic ecosystems is being conducted worldwide (Nõges, Nõges & Laugaste, 2003; Adrian, Wilhelm & Gerten, 2006; Lewandowska & Sommer, 2010; Zohary, Flaim & Sommer, 2020). However, few studies have been conducted on changes in aquatic ecosystems in the natural conditions of North Asia due to fluctuations in the water levels of lakes. The research is mainly focused on anthropogenic impacts or some species of aquatic organisms (Leira & Cantonati, 2008).

The lake’s mountainous conditions and its fluctuating continental subarctic climate affect the physical and chemical characteristics of its ecosystem. The nutrient supply for phytocenoses largely depends on the ecological systems and on the interaction of the catchment-lake system (Carpenter et al., 1998; Bornette & Puijalon, 2011; Nõges et al., 2011; Paerl, Hall & Calandrino, 2011). Nutrient runoff is affected by changes in the amount and dynamics of precipitation (Freeman et al., 2001; Tranvik & Jansson, 2002).

The concentration of nutrients, including ammonium forms of nitrogen, is one of the indicators of anthropogenic pressure on the catchment, especially when considering its agricultural applications (Bornette & Puijalon, 2011; Wang et al., 2014). Research has shown that in 2011–2018, the concentration of ammonium nitrogen decreased, indicating a decrease in the flow of water from the catchment area. In dry years, the PO/COD ratio is below 50%, which indicates the autochthonous origin of organic matter. Many studies have shown the link between inorganic phosphorus and iron, which binds inorganic P in the humic acid complex (Paludan & Jensen, 1995; Paerl, Hall & Calandrino, 2011; Chen et al., 2018). The oxidation–reduction potential (ORP) of Lake Arakhley’s ecosystem ranges from 200 to 218 mV, which creates oxidative conditions and the presence of a number of elements in the highest form of valence (including Fe3+). In high-water years, the iron content ranged from 0.10 to 0.35 mg/L−1 (Usmanov & Zhilin, 2002). In dry years, its concentration decreased by almost as twice compared to high-water years (Table 3). A decrease in the iron content led to a decrease in the content of inorganic phosphorus (Fig. 4). The decrease in the level of nitrogen and phosphorus concentrations in dry years was studied by Coppens et al. (2016). Alterations in the lake’s nutrient concentrations were related to precipitation and inflow volumes.

The findings from 2017–2018 indicate the complexity of the structure and high diversity of the phytoplankton community in the dry period. Phytoplankton biodiversity indices were not calculated during the following periods: 1966–1969, 1970–1971, 1988 and 1994–1995, 2008–2009. In 1966–1976, there were 144 forms of algae identified in the species composition (Morozova, 1981; Morozova & Shishkin, 1973); in 1988 and 1994–1995 there were 103 taxa identified (Ogly, 1993; Ogly, 1995), in 2008–2009 there were 110 taxa identified (Tashlykova & Koryakina, 2013). There were 97 taxa recorded at the rank below genus in the spring and summer of 2017 (Tashlykova, 2019). Taxonomic diversity was determined by diatoms, green, golden, and cyanobacteria as reported in previous studies (Morozova, 1981; Morozova & Shishkin, 1973; Tashlykova & Koryakina, 2013). The taxonomic structure and abundance of phytoplankton species remained at the levels of the 1960s, 1970s, and 2000s in agreement with the ecological modulation.

Phytoplankton’s seasonal dynamics are determined by the dynamics of hydrometeorological conditions. In 2017–2018, two abundance and biomass peaks were distinguished in phytoplankton development, with the first being in the summer. This was the most pronounced due to the development of three groups of algae: green, diatoms, and cyanobacteria. The excessive development of cyanobacteria in the central zone and the lake was not observed despite its low level. The second period in autumn was poorly-expressed due to the development of diatoms, which was also noted in previous studies. However, in the summer of 1970–1989 (Morozova, 1981; Ogly, 1993; Ogly, 1995) cyanobacteria increased in number and biomass.

Notably, research on Lake Arakhley beginning at 1966 has shown that a high phytoplankton biomass corresponds with periods of low-water levels and may be due to the size of the dominant algae species.

The data collected on the phytoplankton structure of the lake is comparable to that of other lakes (Grabowska et al., 2014; Liu et al., 2015 etc). Similar findings on the significance of abiotic factors were obtained through laboratory analyses and stationary research in other water bodies (Bierman & Dolan, 1981; Watson, McCauley & Downing, 1997; Nalewajko & Murphy, 2001; Tian et al., 2013; Chang et al., 2013; Wang et al., 2015; Yang et al., 2016; Yang et al., 2014b; Yang et al., 2017 etc). The condition of phytoplankton communities reflects the reaction of the species to such environmental changes as mixing, water heating, catchment, and physical and chemical properties causing changes in algo- and zoocenoses. These changes have been confirmed by the RDA analysis of the phytoplankton composition and abiotic factors.

Fluctuations in the water-level are one of the core factors affecting the biomass, diversity, composition, and structure of vegetation. Variations in environmental factors including light, oxygen, temperature, and nutrients impact plant growth and germination (Nõges & Nõges, 1999; Geest et al., 2005; Yang et al., 2014a). Fluctuations in the water-level impact aquatic vegetation by altering their amplitude and dynamic regime. Various macrophytes react differently to fluctuations in the water level.

Macrophyte community struction in 2017–2018 and its long-term variations are largely due to a decrease in the water-level of the lake. Different macrophyte species of identical life forms may react differently to fluctuations in the water level (Tan et al., 2020). C. demersum became more abundant when the water level fluctuated (Wang et al., 2016). The lake’s vegetable populations are typically formed under abiotic factors (Hill, Keddy & Wisheu, 1998). A review of the literature on the impact of climate change on aquatic vegetation shows that an increase in temperature affects the growth and development of some plant species (Hossain et al., 2017). However, the increase in temperature had little effect on macrophytes compared to the effect of nutrients (Feuchtmayr et al., 2007; Bornette & Puijalon, 2011). According to Forsberg (1964); Forsberg (1965), when the Ptotal content is higher than 0.02 mg L−1, the growth of Charophyta is inhibited. Analyzing the phosphorus concentration in Lake Arakhley reveals that there was an increase in the phosphorus content up to 0.15 mg L−1 in 2011, which led to the disappearance of the deep-sea species of the Charophyta and mosses.

Conclusion

Cyclic fluctuations in water levels transform the composition of biogenic substances. This, in turn, affects the composition of the dominant complex of the phytocenoses. In dry years, L. comta was dominant in the phytoplankton community, while A. formosa prevailed during high-water years. The hydrophyte community was monodominant and represented by C. demersum populations in dry years. During these periods, macrophyte growth was observed in shallower areas, which were overgrown and lacked complexity. Thicket density and the community diversity in the shallow zone also increased. The lake’s vegetation cover tended to be more diverse during high-water years.

RDA analysis of the correlation between abiotic factors and biocenosis was conducted to determine the parameters of the dry season. Our results indicated that the physical and chemical parameters of water (TDS, pH, EC, ORP, Turb, and Sal), biogenic elements (nitrites, ammonium, and phosphates), depth, and transparency were the most influential abiotic factors leading to the change biodiversity of Lake Arakhley.

Supplemental Information

Supplemental Information 1 Dataset

Click here for additional data file.

Additional Information and Declarations

Competing Interests

Author Contributions

Data Availability

The authors declare there are no competing interests.

Gazhit Ts. Tsybekmitova conceived and designed the experiments, analyzed the data, prepared figures and/or tables, authored or reviewed drafts of the paper, and approved the final draft.

Larisa D. Radnaeva and Arnold K. Tulokhonov conceived and designed the experiments, analyzed the data, authored or reviewed drafts of the paper, and approved the final draft.

Natalya A. Tashlykova, Balgit B. Bazarova and Marina O. Matveeva performed the experiments, prepared figures and/or tables, and approved the final draft.

Valentina G. Shiretorova conceived and designed the experiments, performed the experiments, prepared figures and/or tables, and approved the final draft.

The following information was supplied regarding data availability:

The raw measurements are available in the Supplemental File.

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
