# Peer review of "Phytocenosis biodiversity at various water levels in mesotrophic Lake Arakhley, Lake Baikal basin, Russia"

_PeerJ, doi:10.7717/peerj.11628_

## Round 0.1 · original submission · Major Revisions

Two reviewers have carefully read and evaluated your manuscript. While they both find, that your findings would be of interest to other scientists they also raise severe criticisms. Please find their comments and suggestions further down in this letter.

First of all, it is of uttermost importance that the study is based on original data, that the dataset has not been published before. As pointed out by one of the reviewers this is not the case. The authors must therefore clarify this aspect before a revised manuscript can be submitted.

Both reviewers and I find the linguistics and writing style problematic to such an extent that it is difficult to understand the meaning. It is highly recommended that the manuscript is proof-read by a native English speaking scientific person.

Overall the manuscript appears rather unfinished and the authors must pay attention to make the text more clear and readable as well as correct flows and add the missing information. Again, both reviewers have provided good guidance and all their comments and suggestions must be addressed.

Reviewer 1 ·

Basic reporting

The article is written in English and the language should be checked by English native spikier. Literature references are sufficient to demonstrate the work fits in to broader field of knowledge. The weak point that 28 – 29% of referred literature are in Russian. The structure of article is according the standards of PeerJ. Figures and titles of them must be revised, for example Figure 6 is a copy from other article or report. Also supplementary data should be revised, descriptions of data are in Russian and is not clear from with years data are. There is no any reference to the supplementary data in the text of article. There is no define the hypothesis of the research improved with included results.

Experimental design

Research questions and knowledge gaps are defined and clear. Methods have not described sufficiency to be reproducible for other investigators.

Validity of the findings

Data used in this article have been partly published in other articles. What is a novelty of this research? It would be good to see the full version the article “The Effect of Climatic Shifts on Biodiversity of Phytocenosis: Lake Arakhley (Eastern Siberia, Russia)”, International Journal of Ecology & Development (ISSN 0973-7308 (Online)), 2020, Volume 35, Issue Number 3, 77-90” by the same authors where the findings described in the abstract seem exactly the same as in this submitted article.

Additional comments

The aim and data of this research is actual in context of global situation and climate changes in the world. The most data of the mesotrophic Lake Arakhley are available in publications in Russian thus reducing their usability for wider public and should be published in in the international scientific journal. Questionable is a novelty of this research since the same findings can be found in other already published article and there is no reference to it. Despite the fact that the structure of submitted article is according roles of journal the text in chapters must be revised. Please, see more detailed comments and suggestions with attached pdf file.

Annotated reviews are not available for download in order to protect the identity of reviewers who chose to remain anonymous.

Reviewer 2 ·

Basic reporting

Dear authors, thank you for an interesting article. Data provided by authors definitely supplement present knowledge on phytocenoses from aquatic ecosystems. An abstract is clearly written, although I would suggest to start with the sentence " the purpose of this study..." and/or re-organize the text a bit since sentences "this paper presents..." and "the purpose of this study..." are repetitive.
For the field background/context provided I found it too general, hence more baseline/recent examples could be included describing water level fluctuation study cases.
Please check if figures references in the text are corresponding appropriate figures (e.g. line 290 - there should be figure 7 instead of figure 6), this should be checked throughout the text. Figure 7 needs more explanation/description as to abbreviations used in the figure. Figure 6 - translate y axis from Russian to English.
Line 367 - WLF abbreviation is used. Since it appears just twice and just at the end of the article, I do not find it appropriate form writing style point of view. If this is an important key-word/abbreviation, why not to include it already at the beginning of the article/abstract/key-words.
The main objections however are concerning language and accuracy throughout the text, the sentences are not always clear and easy to understand, what makes it difficult to read and follow the story written. There are writing style differences between literature reference paragraphs and text written by the authors. Line 314, please check if “ultra-continental” is the appropriate expression.
As to accuracy, more comments are given further in other review chapters. E.g. there is extensive amount of raw data given in the supplemental files, still, tables are without captions and descriptions, there are no references given supporting data, neither I found where in the main text authors are referring to supplemental material. Moreover, supplemental tables are not completely translated, column names contain information written in Cyrillic script what will be difficult to understand for future readers. I got impression manuscript has not been ready for submitting, e.g. lines 341 - 354 - should be rewritten to correct writing style, organize text in more comprehensive way, more appropriate when it comes to Latin species name writing (genus names can be abbreviated after first use, however you have to decide on writing style and keep it throughout the text).

Experimental design

The research question is well defined and meaningful. Although study is more descriptive, it still fits well the scope and aim of the journal.
Nevertheless, methods description needs to be improved, it is not clear where exactly in the lake (coordinates or to show in the map) samples were taken from, when (time of the year), how deep were surface level samples (down to 0.5 m or?) and what is a depth of lake bottom sampling (sediments? just above bottom level or?). For hydrochemistry since you are referring to unpublished data, please give more detailed information about methods used - devices, frequency, analytical methods if any etc.
Line 127 - please correct Cyrillic script to Latin alphabet.
Lines 134 - 135 "some...parameters ...were measured with...." - please be more precise -what parameters?
Lines 139 - 140 - reference for Hansen method is not given.

Validity of the findings

The findings are well supplementing existing knowledge of the field. Descriptive chapters are rather long, hence it could be worth considering to present them in the tables and/or figures (this is just a suggestion), e.g. lines 208 - 221, 243 - 252.
Please support your findings by linking/comparing it more to other studies done so far, e.g. lines 355 - 365, this paragraph can be expanded. Since water level changes are the main research question, I would suggest to re-organize discussion part and start with a paragraph on water level fluctuations in order to support results.

Additional comments

I would like to encourage authors to elaborate/correct manuscript and submit again after major revisions.

---

## Round 0.2 · Major Revisions

I agree with the reviewer, who kindly re-reviewed the manuscript, that the authors have made very good progress with the manuscript. They followed much of the advice and have in most cases explained well which changes they made.

However, the reviewer still raised some points that need to be addressed (please see the annotated document).

In addition, the number of figures and tables is a bit overwhelming and especially it should be considered whether all the tables are needed in the main part or better could appear in the supplementary materials.

Finally, it would be strongly advised to run yet another linguistic proof-reading.

Despite this requires another round of editing, I hope you agree that it all leads to a valuable scientific contribution.

Reviewer 1 ·

Basic reporting

The authors have made significant improvement to the manuscript (MS). They have removed from the references a number of Russian untranslated publications, revised Figures and titles. And after revision is clear the purpose and hypothesis of the MS.

Experimental design

Methods have supplemented with additional text, references and now are clear for readers.

Validity of the findings

The validity of the data and findings are explained and proofed with supplement copy of the asked Paper.

Additional comments

Thank you! You have made a great job, your study is very interesting and this article will be an important international contribution and complement to environmental research, especially for the Lake Arakhley. Anyway I have found some things should be pay your attention, please find them in the separate attached file.

Annotated reviews are not available for download in order to protect the identity of reviewers who chose to remain anonymous.

---

## Round 0.3 · accepted · Accept

Your kind responses to the comments/suggestions during the second round are recognized and I believe your manuscript now deserves to be published.